# Studies on the Oxidative Damage of the Wobble 5-Methylcarboxymethyl-2-Thiouridine in the tRNA of Eukaryotic Cells with Disturbed Homeostasis of the Antioxidant System

**DOI:** 10.3390/ijms252212336

**Published:** 2024-11-17

**Authors:** Malgorzata Sierant, Rafal Szewczyk, Agnieszka Dziergowska, Karolina Krolewska-Golinska, Patrycja Szczupak, Przemyslaw Bernat, Barbara Nawrot

**Affiliations:** 1Department of Bioorganic Chemistry, Centre of Molecular and Macromolecular Studies, Polish Academy of Sciences, Sienkiewicza 112, 90-363 Lodz, Poland; karolina.krolewska-golinska@cbmm.lodz.pl (K.K.-G.); patrycja.szczupak@cbmm.lodz.pl (P.S.); barbara.nawrot@cbmm.lodz.pl (B.N.); 2LabExperts, Sokola 14, 93-519 Lodz, Poland; rafal.szewczyk@labexperts.com.pl; 3Institute of Organic Chemistry, Faculty of Chemistry, Lodz University of Technology, Zeromskiego 116, 90-924 Lodz, Poland; agnieszka.dziergowska@p.lodz.pl; 4Department of Industrial Microbiology and Biotechnology, Faculty of Biology and Environmental Protection, University of Lodz, Banacha 12/16, 90-237 Lodz, Poland; przemyslaw.bernat@biol.uni.lodz.pl

**Keywords:** tRNA damage, desulfuration, oxidative stress, LC-MS/MS analysis, 5-methylcarboxymethyl-2-thiouridine, mcm5S2U, 5-methylcarboxymethyl-4-pyrimidinone riboside, mcmH2U, 5-methylcarboxymethyl-uridine, mcm5U

## Abstract

We have previously shown that 2-thiouridine (S2U), either as a single nucleoside or as an element of RNA chain, is effectively desulfurized under applied in vitro oxidative conditions. The chemically induced desulfuration of S2U resulted in two products: 4-pyrimidinone nucleoside (H2U) and uridine (U). Recently, we investigated whether the desulfuration of S2U is a natural process that also occurs in the cells exposed to oxidative stress or whether it only occurs in the test tube during chemical reactions with oxidants at high concentrations. Using different types of eukaryotic cells, such as baker’s yeast, human cancer cells, or modified HEK293 cells with an impaired antioxidant system, we confirmed that 5-substituted 2-thiouridines are oxidatively desulfurized in the wobble position of the anticodon of some tRNAs. The quantitative LC-MS/MS-MRMhr analysis of the nucleoside mixtures obtained from the hydrolyzed tRNA revealed the presence of the desulfuration products of mcm5S2U: mcm5H2U and mcm5U modifications. We also observed some amounts of immature cm5S2U, cm5H2U and cm5U products, which may have indicated a disruption of the enzymatic modification pathway at the C5 position of 2-thiouridine. The observed process, which was triggered by oxidative stress in the living cells, could impair the function of 2-thiouridine-containing tRNAs and alter the translation of genetic information.

## 1. Introduction

Transfer RNAs (tRNAs) are important molecules in all living organisms that translate messenger RNA (mRNA) and deliver amino acids to the ribosome in the order specified by genetic information. The biology of tRNA maturation and the principles of its primary function in translation are well understood and belong to so-called rudimentary knowledge. However, tRNA has once again become the important research target due to advances in new technologies, particularly new methods based on mass spectrometry, RNA sequencing, and NMR spectroscopy. These new technologies offer the opportunity to expand the knowledge of new, unknown modifications in tRNA or in other RNA types, to follow the cellular dynamics of the RNA modification profiles triggered in particular by stress, and to discover the cellular pathways and enzymes responsible for the introduction or removal of nucleoside modifications in RNA [1,2,3,4,5,6]. Our current knowledge of the enzymes that modify nucleosides in RNA is still incomplete, e.g., about 23% of human enzymes have not yet been identified and another 20% require further experimental validation [7]. This is why it is so important to advance research in this area.

TRNAs are the most abundant category of small non-coding RNA molecules in the cell and have the highest modification rates, which distinguishes them from other types of cellular RNAs. To date, about 170 different nucleoside modifications have been identified in RNA, and at least 120 modified nucleosides are found in tRNAs [8,9,10,11,12]. Modifications in tRNA can be classified according to their function, type, and position within the tRNA structure. Modifications in the core region of tRNA, located within arms D and TΨC, improve and stabilize the tertiary structure of tRNA and maintain the balance between the flexibility and stability of the L-form of the tRNA unit. Nucleoside modifications located in the anticodon–stem–loop domain modulate codon–anticodon interactions and play a regulatory function during decoding [13,14]. Modified nucleosides in tRNAs can also be determinants for other components of the translational apparatus such as aminoacyl–tRNA synthetases (aaRSs) [15,16]. In addition, tRNA modifications can be recognition elements of ribonucleases, leading to the generation of tRNA fragments that influence numerous cellular processes [17].

Sulfur is an important component of various classes of biomolecules such as proteins, nucleic acids, sugars, sulfur-containing vitamins, e.g., thiamine, iron–sulfur cofactors for enzymes, a variety of sulfur-containing metabolites, etc. [18]. In nucleic acids, the sulfur substitution of an oxygen atom on the nucleobase or in the phosphate backbone leads to a variety of sulfur-modified nucleosides or nucleotides [19]. TRNAs also contain sulfur-modified nucleosides, which are incorporated post-transcriptionally into the maturing tRNA chain by specific modifying enzymes. Depending on the position at which the modifications occur in the tRNA chain, they have different functions, such as stabilizing tRNA structures, enabling the identification of tRNAs by aaRSs, improving ribosomal binding to aminoacylated tRNAs, maintaining the reading frame, and ensuring correct codon–anticodon base pairing [20].

In general, sulfur-containing modifications of nucleosides are found at seven different positions in the tRNA chain. These are positions 8, 9, 32, 33, 34, 37 and 54 of the tRNA chain. However, the presence of modifications depends on the organism from which the tRNA originates. The modification at position 9 (S4U_9_) is only characteristic for the archaeon *Thermoplasma acidophilum* [21] and the modification at position 33 (S2U_33_) is characteristic for *Trypanosomatis*. Bacterial and archaeal tRNAs contain five thio-nucleosides: 4-thiouridine at position 8 (S4U_8_), 2-thiocytidine at position 32 (S2C_32_), 5-methylaminomethyl-2-thiouridine, 5-carboxymethylaminomethyl-2-thiouridine or their 5-(c)mnm-*S*-geranyl-2-thiouridine derivatives at position 34 (mnm5S2U_34_, cmnm5S2U_34_, (c)mnm5geS2U_34_), and 2-methylthio-*N*6-isopentenyladenosine, 2-methylthio-*N*6-hydroxyisopentenyladenosine or 2-methylthio-*N*6-threonyl-carbamoyl adenosine at position 37 (mS2i6A_37_, m5S2io6A_37_, mS2t6A_37_) [22]. Some of the thermophilic bacteria have an additional 5-methyl-2-thiouridine (m5S2U_54_) at position 54. Eukaryotic organisms contain sulfur-modified nucleosides at two positions—34 and 37 of the anticodon loop. There are 5-methylcarboxymethyl-2-thiouridine (mcm5S2U_34_) and derivatives that differ by the substituent at C5 and 2-methylthio-*N*6-hydroxyisopentenyladenosine (m5S2io6A_37_) or 2-methylthio-*N*6-threonyl-carbamoyladenosine at position 37 (mS2t6A_37_) [23,24].

5-Substituted 2-thiouridines (R5S2U) are universal thio-modifications found in three domains of life. The modified R5S2U contains a sulfur atom instead of an oxygen atom at position 2 of the uracil ring and an additional substituent at position 5, the nature of which depends on the origin of the tRNA. Depending on the organism and subcellular localization of the tRNA, the substituent R5 can be hypermodified to methylaminomethyl-(mnm-) and carboxymethylaminomethyl- (cmnm-) in bacteria and archaea, to methylcarboxymethyl-(mcm-) in the eukaryotic cytosol or to taurinomethyl-(τm-) in the mitochondria of mammals [19,24,25]. Several functions have been proposed for R5S2U_34_ modification. The first important function, based on the localization at the wobble position of the tRNA, is the recognition of the third nucleoside of the codon triplet in codon–anticodon interaction. R5S2U units primarily facilitate Watson–Crick base pairing with adenosine at the 3′-end of the 5′-NNA-3′ mRNA codons and restrict wobble base pairing with G at the third position of the 5′-NNG-3′ codons. The other function, the sulfur atom at position C2 of the uracil ring, is the identity-promoting element in aminoacylation reactions [26] and increases translation efficiency at the ribosome by increasing the binding affinity of aminoacylated tRNAs to the A site of the ribosome as well as the rate GTP hydrolysis [27]. In addition, the R5S2U_34_ modification preserves translational fidelity by preventing +1 and +2 ribosome frame shifting [28,29]. In the yeast *Saccharomyces cerevisiae*, mcm5S2U occurs mainly at the wobble position of the tRNA specific for Lys, Glu, or Gln (tRNA^Glu^, tRNA^Lys^ and tRNA^Gln^). As mentioned above, 5-substituted S2U is also found in human mitochondrial lysine specific tRNA (hmttRNA^Lys^).

As shown by Corsaro and Pistara in the late 1990s nucleosides containing the thiocarbonyl group are sensitive to oxidizing conditions despite their stabilizing properties [30]. The 5-substituted 2-thiouridines have been the subject of interest of our group (researchers from CMMS PAS and TUL) for several years [31,32,33,34,35]. Studying this modification, we discovered that the 2-thiouridine (S2U), alone or incorporated into an RNA oligonucleotide chain, was desulfured under oxidative conditions, i.e., the sulfur atom was removed from the S2U molecule, and the two reaction products deprived of the sulfur atom were obtained: 4-pyrimidinone riboside (H2U) and uridine (U), Figure 1 [31].

The ratio of the S2U desulfuration products depended on the reaction conditions that is: pH, the type of R5 substituent, and the type and concentration of the oxidizing reagent. The reaction carried out at the lower pH (6.6) produced mainly H2U (80% H2U and 20% U), while in the reaction carried out at the higher pH (7.6), the main product was uridine (80% U and 20% H2U) [31,32]. In addition, we have shown that the substituent R5, although not directly involved in the reaction with the oxidizing agent, strongly affects the pathway of desulfuration and the product ratio. When substituent R5 was changed, the amount of R5H2U decreased in the following order: H > CH_3_O- > CH_3_OC(O)CH_2_- > HOC(O)CH_2_NHCH_2_- ~ CH_3_NHCH_2_. This effect was observed within the physiological pH range (pH between 6.6 and 7.6), but was more pronounced at lower pH values (pH 6.6) [34]. Conformational analysis of the R5H2U desulfuration product showed that R5H2U predominantly adopts the C2′-*endo* conformation regardless of the nature of the R5 substituent, in contrast to the C3′-*endo* conformation of R5S2U [31,35]. Due to the altered pattern of hydrogen bonding, the nucleoside H2U forms a different base pairing with the nucleoside in the complementary RNA chain than S2U. 2-Thiouridines hybridize preferentially with adenosines (S2U-A), whereas wobble pairing with guanosine residues (S2U-G) is restricted due to the less effective hydrogen bonding between the N1H donor of guanine and the sulfur acceptor in 2-thiouracil. After the conversion of S2U to H2U, the situation is different. The H2U nucleoside hybridizes preferentially with guanosine because it has a different pattern of hydrogen bonds than the acceptors and donors of uridine and 2-thiouridine [35]. These results suggest that the conversion of R5S2U within the tRNA wobble position to its R5H2U product may be of great biological significance because the structural and functional features of the R5H2U unit are highly altered compared to the original R5S2U.

In the present study, we addressed the question of whether the desulfuration of RS2U is possible in the natural tRNAs in living cells exposed to oxidative stress. To solve this issue, we used eukaryotic model cells with impaired antioxidant systems: the yeast *Saccharomyces cerevisiae*, human cancer cells and modified HEK293 cells with mutations in the *SOD2*, and *Cat* genes that resulted in the suppression of the expression of catalase (Cat) and located in mitochondria manganese superoxide dismutase (MnSOD2), enzymes important for the cellular antioxidant system. To induce oxidative stress in the cells, we used stressors known from the literature: H_2_O_2_, NaAsO_2_ and NaClO [36]. First, we determined the maximum concentrations of oxidizing reagents that can be applied in cell culture and do not lead to cell death, and we created cell culture conditions under oxidative stress. Subsequently, the tRNAs extracted from the cells cultured under oxidative stress were hydrolyzed to nucleosides and subjected to LC-MS/MS analysis to search for mcm5S2U-tRNA desulfuration products.

## 2. Results and Discussion

### 2.1. Characteristics of Cells Used in Research

In the first part of the studies, we used three laboratory strains of yeast *Saccharomyces cerevisiae* (1) INVSc1, the commercial, fast-growing, diploid strain designed primarily for protein expression [*MATa his3D1 leu2 trp1-289 ura3-52 MAT his3D1 leu2 trp1-289 ura3-52*], (2) M3, the wild-type strain [*MATa lys2 his4 trp1ade2 leu2 ura3*], and (3) M3Δ*sod1*, the isogenic-to-M3 parent strain [*MATa lys2 his4 trp1ade2 leu2 ura3 Δsod1::kanMX4*] with deletion of the copper–zinc superoxide dismutase gene. Yeast is a very useful and practical model microorganism for tRNA research because its culture is simple, inexpensive, and allows suitable amounts of biomass to be obtained in a short time. The yeast *S. cerevisiae* contains three types of tRNA with the modification mcm5S2U at the wobble position of the anticodon. These are tRNA^Lys^, tRNA^Glu^ and tRNA^Gln^, which represent a useful model for the study of intracellular R5S2U desulfuration. A description of yeast strains and culture conditions can be found in Appendix A.

In the next experiments, we examined human cancer cells, both adherent and growing in suspension, derived from eight human cancer cell lines (HeLa, K562, MOLT-4, A431, A375, A549, U87 MG, MCF-7) and the human HEK293 cell line as a model of normal cells. A description of all the cell lines and culture conditions can be found in Appendix A. Cancer cells served as the model cells with intracellular oxidative stress [37,38]. Cancer cells usually show the reduced activity of antioxidant enzymes compared to normal cells. The reduced activity of superoxide dismutases (Cu-ZnSOD1, MnSOD2) and catalase (Cat), which is found in many types of cancer cells, leads to the accumulation of ROS (reactive oxygen species) in the cells, which in turn leads to intracellular oxidative stress and its consequences for biomolecules.

The third cell type used in the studies were HEK293 cells that were genetically modified using the CRISPR/Cas9 gene editing system to damage the human *SOD1*, *SOD2*, and *Cat* genes.

#### 2.1.1. Application of the CRISPR/Cas9 Gene Editing System for the Genome Modification of Human HEK293 Cells

CRISPR (clustered regularly interspaced short palindromic repeats) is an effective and precise system for genome editing in living cells that is used in many scientific fields. We used the CRISPR/Cas9 system to introduce mutations in the human genes for catalase (Cat) and cytoplasmic (Cu-Zn-SOD1) or mitochondrial (Mn-SOD2) superoxide dismutases to inhibit their expression in the studied cells. These three enzymes are an important part of the human antioxidant system, which is responsible for neutralizing reactive oxygen species in the cells and protecting the cells from oxidative stress and free radicals. Superoxide dismutase catalyzes the conversion of the toxic superoxide anion O_2_^•−^ into oxygen and hydrogen peroxide. Catalase is responsible for the decomposition of hydrogen peroxide into water and oxygen.

Edit-R, a commercial predesigned, synthetic, sgRNA, was used, with three variants per gene, targeting three human genes: *Cat*, catalase gene [GenBank: AY545477, target sequence: sg1: 16,228–16,247, sg2: 15,200–15,219, sg3: 24,427–24,446]; *SOD1*, superoxide dismutase 1 gene [GenBank: AY835629, target sequence: sg1: 7730–7749, sg2: 139–158, sg3: 4248–4267]; and *SOD2*, superoxide dismutase 2 gene [GenBank: AY26790, target sequence sg1: 528–547, sg2: 476–495, sg3: 452–471], with the Edit-R sgRNA targeting the human *PPIB*, peptidylprolyl isomerase B gene [GenBank: AY962310,] as a positive control for gene editing, TrueCut Cas9 v2 nuclease (5 µg/µL), and Lipofectamine CRISPRMAX transfection reagent. The sequences of the Edit-R sgRNAs and PCR primers are listed in Appendix A. After transfection, the culture of the HEK293 cells was continued under optimal conditions according to the protocol described in Materials and Methods. After the next 3 days, cells were harvested, counted, and diluted in fresh medium to the density of 1 cell per 100 µL and transferred into 96-well plates to generate single-cell clones. After the next 3 weeks of growth, the first selection of cells was performed and those cells in the wells of the 96-well plate were selected that showed characteristics of a single cell origin, i.e., they grew in a single location, not scattered over the entire surface of the well. These cells were selected and transferred to 48-well plates, then to 24-well plates, then to 6-well plates and finally to culture flasks. Appendix A provides a simple calculation of the efficiency of the CRISPR/Cas9 procedure and clone selection.

The initial selection of the clones obtained was performed via the Western blot method using primary antibodies specific for SOD1, SOD2 or Cat antigens, to detect cells not expressing the target genes. For the HEK293*ΔSOD1* cells, 96 clones were tested, all of which expressed the *SOD1* gene (0 positive clones, SOD1-sgRNA/Cas9 efficiency 0%). For the HEK293*ΔSOD2* cells, 109 clones were tested; 5 of the clones did not express the *SOD2* gene (SOD2-sgRNA/Cas9 efficiency 4.6%). For the HEK293*ΔCat* cells, 20 clones were tested; 19 clones did not express the *Cat* gene (Cat-sgRNA/Cas9 efficiency 95%), the data for which are summarized in Appendix A. Figure 1 shows the Western blotting analysis of HEK293*ΔSOD1*, HEK293*ΔSOD2*, and HEK293*ΔCat*positive clones. The positive clones selected in this way were then subjected to a T7 endonuclease assay and genomic DNA sequencing, which confirmed the presence of mutations in the target genes resulting from the CRISPR/Cas9, as seen in Appendix A. As the result of these experiments, we selected 5 clones of HEK293 cells with the deletion of the *SOD2* gene (HEK293*ΔSOD2*) and 19 clones with the deletion of the *Cat* gene (HEK293*ΔCat*). Unfortunately, despite our efforts, all the clones with suspected damage to the *SOD1* gene continued to express SOD1 protein, which was confirmed by the immunoblotting method with antibodies specific for SOD1 antigen. Therefore, the subsequent R5S2U desulfuration studies were performed without HEK293*ΔSOD1* cells, which were not available at that time.

### 2.2. Optimization of Yeast Cell Culture Conditions Under Oxidative Stress

#### 2.2.1. Evaluation of Yeast Viability in the Presence of Oxidative Stress-Inducing Reagents

The yeast (InvSc1, M3 and M3*Δsod1* strains) were cultured under optimal growth conditions until they reached an exponential growth phase (OD_600_ = ~0.6–0.8). In this growth phase, the cells were healthiest, divided intensively, and the number of cells increased rapidly. The yeast cells were exposed to oxidizing agents such as H_2_O_2_ (at concentrations of 0, 5, 10, 25, 50, 75, 100 mM), NaAsO_2_ (at concentrations of 0, 10, 20, 40, 60, 80, 100 mM), or NaClO (at concentration of 0, 3, 4, 5, 7, 10 mM) for the 1 or 2 h, to see how they grew under stress conditions and to determine the range of oxidant concentrations in which complete cell damage and cell death did not occur. The following tests were performed to evaluate the viability of the yeast cells in the presence of oxidizing agents: the spotting test and the colony-forming unit (CFU) test. In addition, live and dead cell staining tests were initially performed (methylene blue staining or propidium iodide with fluorescein diacetate staining), but these tests were abandoned because strains M3 and M3*Δsod1* were only slightly stained. Figure 2 shows the example of the spotting test used to evaluate the sensitivity of the tested yeast strains to hydrogen peroxide. The remaining results can be found in Appendix A. The detailed protocol for the yeast viability assays can be found in Materials and Methods. The so-called “safe concentrations” of oxidizing reagents at which yeast viability was at least 50% are listed in Table 1. The determined conditions were used in further studies.

#### 2.2.2. Determination of the ROS Level in Yeast Cells After Incubation with H_2_O_2_, NaAsO_2_ or NaClO

The yeast cells were grown under optimal conditions until the exponential growth phase was reached (OD_600_ = ~0.6–0.8). For the measurement of intracellular ROS, the 2,7-dichloro-dihydrofluorescein diacetate (DCFDA) assay was performed with a non-fluorescent precursor of 2,7-dichloro-dihydrofluorescein. Detailed instructions can be found in Materials and Methods. After the cells reached the appropriate density, they were incubated with the oxidizing compounds for 1 or 2 h, washed with PBS buffer, and the fluorescence of the samples was measured using a plate reader. PBS buffer and untreated yeast cells served as a negative control (no ROS). The level of ROS was measured in whole, intact cells (which were not lysed) suspended in PBS buffer. The results of the ROS analysis are shown in Figure 3 and Appendix A. The results show that reactive oxygen species (ROS) are formed in yeast cells under the influence of the oxidizing agents used. The highest concentration of ROS was observed in yeast cells exposed to H_2_O_2_ or NaAsO_2_.

The ROS level induced by NaClO was lower (1–2%) in both of the yeast strains (M3 and M3*Δsod1*), as seen in Appendix A. Care should be taken to read the counted ROS within the so-called “safe concentrations” of oxidizing agents, as seen in Table 1. The significant increase in ROS concentration visible in both of the graphs in Figure 3 means that the yeast cells were damaged by the oxidizing agents used at higher concentrations (H_2_O_2_ > 25 mM, NaAsO_2_ > 40 mM).

#### 2.2.3. Monitoring the Reduction in the Amount of mcm5S2U-tRNA in Yeast Treated with H_2_O_2_, the γ-toxin Assay

We confirmed the reduction in mcm5S2U-tRNA with the mcm5S2U modification in the wobble position using a γ toxin assay [39]. Gamma toxin from the yeast *Kluyveromyces lactis* is an enzyme that specifically recognizes the mcm5S2U in the wobble position of tRNA and cleaves the tRNA at this site (between nucleosides at the 34 and 35 position in the anticodon loop). If mcm5S2U is desulfured, it is not recognized by γ-toxin and is not hydrolyzed. The recombinant γ-toxin was overexpressed in the bacterial strain BL21(DE3) and purified by affinity chromatography on Ni-NTA agarose according to the protocol described in Materials and Methods, yielding several milligrams of active enzyme. For the experiments, we used the yeast *Saccharomyces cerevisiae*, INVSc1 the commercial strain, which was the most resistant to H_2_O_2_ compared to the other two strains (IC_50_ = 50 mM, Table 1). Total cellular RNA isolated from yeast exposed to oxidative stress (H_2_O_2_) during culture was reacted with γ-toxin. The hydrolysis of mcm5S2U-tRNA was monitored via Northern blot hybridization of the resulting full-length tRNA and tRNA cleavage products with a complementary DNA probe labeled with the ^32^P-phosphate group at the 5′ end. As a result of the assay, two bands were observed on the membrane: the band corresponding to the full-length tRNA (tRNA that does not contain mcm5S2U modificationis not recognized, and not hydrolyzed by γ-toxin), and the second lower band, corresponding to the half fragment of the mcm5S2U-tRNA (after hydrolysis by γ-toxin), as seen in Figure 4B. 25SrRNA was used as a reference RNA.

The second assay used to quantitatively assess the reduction in mcm5S2U-tRNA was qRT-PCR with specific primers (shown schematically in Figure 4A). 25S rRNA, which did not contain mcm5S2U modification, was used as a negative control for the tRNA hydrolysis reaction catalyzed by γ-toxin, as seen in Figure 4C.

Based on the results of qRT-PCR, we estimated that the use of H_2_O_2_ (up to 50 mM) during the culture of yeast resulted in a loss of approximately 80% of the input mcm5S2U-tRNA^Glu^, which could indicate oxidative desulfuration of the mcm5S2U-tRNA or the disruption of cellular pathways (tRNA-modifying enzymes) specific for the introduction of mcm5S2U modification into the wobble position of the anticodon of tRNA^Glu^ via oxidative stress. After the application of 100 mM H_2_O_2_ during yeast culture, we observed almost 100% loss/damage of the mcm5S2U-tRNA, but this result was most likely due to yeast cell death and the direct effect of hydrogen peroxide on tRNA. This result was confirmed by the Northern blot, which allowed the identification of uncleaved tRNA and mcm5S2U-tRNA^Glu^ cleavage product by the γ toxin. Another interesting result was found in the cells that were not treated with H_2_O_2_ during culture. About 40% of the analyzed tRNA^Glu^ from the not-treated yeast was not hydrolyzed by the γ toxin, which may indicate that this tRNA also did not contain the mcm5S2U modification at the wobble position of the anticodon, as seen in Figure 4C.

### 2.3. Optimization of Culture Conditions for Human Cancer Cells Under Oxidative Stress: Evaluation of Human Cells Viability and Intracellular ROS Level

Human cancer cells were subjected to exogenous oxidative stress via exposure to H_2_O_2_, NaAsO_2,_ or NaClO stressors during culture. The MTT assay was performed to evaluate the cell viability under oxidative stress conditions and IC_50_ values were determined for all the cells tested, as seen in Table 2, Figure 5 and Appendix A.

The HEK293 cells (IC_50_ = 14.5 µM) and their mutant variants HEK293*ΔSOD2* and HEK293*ΔCat* (IC_50_~10 µM) were the most sensitive to oxidizing reagents, especially H_2_O_2_, while the MOLT-4 and K562 suspension cells (IC_50_ > 200 µM) were the least sensitive. The other compounds, NaClO and NaAsO_2_, used at the indicated concentrations, were not as lethal to most of the cells as H_2_O_2_. The IC_50_ determined for NaAsO_2_ was ~200 µM and for NaClO was ~100 µM for almost all the cells tested. These conditions were used for further research.

In addition, we have confirmed for all the oxidizing compounds used—H_2_O_2_, NaAsO_2_ or NaClO—that they cause the formation of ROS in cells (2,7 dichloro-dihydrofluorescein diacetate assay), as seen in Appendix A. When the concentration of the oxidizing reagent increases during incubation with the cells, the level of intracellular ROS also increases. Most ROS are formed in the cells treated with H_2_O_2_; the other compounds cause the approximately 20–80 times lower amount of ROS (corresponding values for NaAsO_2_ and NaClO). The ROS concentration in the cells changes over time, initially increasing and peaking after 1–2 h, and then decreasing again. This phenomenon can be explained by the action of the cellular antioxidant system.

Due to the natural intracellular oxidative stress that exists in cancer cells, we evaluated and compared the ROS levels in the cancer cells that were not treated with oxidizing compounds using the 2,7 dichloro-dihydrofluorescein diacetate assay. For this experiment, we selected HEK293, HeLa, A375, A549, A431, and MCF-7 cells and performed the assay with the same number of cells (25 × 10^3^ cells/well). The cells were incubated with 2,7-dichloro-dihydrofluorescein diacetate for 30 min, washed, and spread on the wells of a 96-well plate, and the fluorescence values were measured after 1, 2, 4 and 6 h of incubation at 37 °C, 5% CO_2_. The fluorescence values obtained for the different cell lines were compared with the fluorescence determined for the HEK293 cells as a baseline value. We found that all the cancer cells tested had some level of intercellular ROS above the level found in HEK293 cells, as seen in Appendix A.

### 2.4. Identification of R5S2U Desulfuration Products by LC-MS/MS (MRMhr) Analysis

#### 2.4.1. R5S2U Desulfuration in Yeast tRNA^Glu^

The yeast tRNA^Glu^ specific for glutamic acid, contains the modified nucleoside mcm5S2U at the wobble position of the anticodon. Based on our previous observations of the desulfuration of 2-thiouridine under oxidative conditions, we decided to verify whether such a desulfuration process occurs in the natural tRNA in yeast cells exposed to oxidative stress during their growth. For this purpose, we used the highly sensitive and highly selective LC-MS/MS (MRMhr) technique (QTRAP 6500+ mass spectrometer coupled to Exion LC System, Sciex, Framingham, MA, USA) and synthetic mcm5S2U, mcm5H2U, mcm5U, cm5S2U, cm5H2U, and cm5U as nucleoside standards for LC-MS/MS analysis, Appendix A. After culture, the yeast (*S. cerevisiae*, M3 and M3*Δsod1* strains) in the presence of hydrogen peroxide (0–100 mM H_2_O_2_) the specific tRNA^Glu^, was isolated from total cellular RNA by binding to the complementary biotinylated DNA probe attached to the streptavidin–agarose resin. The resulting pure tRNA^Glu^ was subjected to simultaneous hydrolysis by two nucleases: Benzonase (endonuclease from *Serratia marcescens*) and Phosphodiesterase I from *Crotalus adamanteus* venom (5′-exonuclease), and subsequently dephosphorylated by alkaline phosphatase CIAP. The resulting nucleoside mixture was analyzed by LC-MS/MS as described in detail in Materials and Methods and in Appendix A. Almost all the nucleosides of interest (mcm5S2U, mcm5H2U, mcm5U, cm5S2U, cm5H2U and cm5U) were detected in the samples analyzed, but the amount of nucleosides identified was dependent on the yeast strain from which the tRNA^Glu^ was derived (M3 or its *sod1* deletion variant, M3*Δsod1*) and on the oxidative conditions to which the yeast was exposed. The most abundant in the nucleosides’ mixture were the 5-substituted 2-thiouridines: mcm5S2U and cm5S2U, but mcm5H2U and mcm5U were also detected, which were probably products of the oxidative desulfuration of mcm5S2U. The cm5H2U nucleoside was only detected in the samples from yeast after the application of 100 mM H_2_O_2_ during growth. However, this result did not indicate desulfuration in the living cells, as the yeast belonging to the M3 and M3*Δsod1* strains did not survive under 100 mM H_2_O_2_ conditions. It can be assumed that the low percentage of identified mcm5H2U and cm5H2U in the other samples (0–25 µM H_2_O_2_ M3; 0–5% M3*Δsod1*) could be due to the sensitivity of the analyzed H2U derivatives and the possible decay in the yeast cells and/or during mass analysis. The percentages shown in the graphs indicate the percentage of the individual modified nucleosides in the nucleoside mixture. 100% was the sum of all the 2-thiouridine derivatives tested.

First, we found certain amounts of mcm5H2U (4% M3; 2% M3*Δsod1*) and mcm5U (11% M3; 30% M3*Δsod1*) nucleosides in the so-called non-oxidatively treated cells, which could be due to the action of intracellular ROS in the yeast cells.

The nucleoside mixture derived from tRNA^Glu^ isolated from the M3 strain, contained mcm5S2U, the content of which decreased from 40% to 31% with increasing H_2_O_2_ concentration (0–25 mM) used during yeast culture. At the same time, small amounts of mcm5H2U (4–5%) and mcm5U (11–21%) were detected. The levels of cm5S2U (41–43%) and cm5U (2–3%) remained fairly constant regardless of the H_2_O_2_ concentration used during culture.

The nucleoside mixture from tRNA^Glu^ isolated from the M3*Δsod1* deletion variant at a safe concentration of 0–5 mM H_2_O_2_ contained a constant level of mcm5S2U (26%), small amounts of mcm5H2U (2%), and a relatively high constant proportion of mcm5U (30%). After exceeding the safe H_2_O_2_ concentration during yeast culture (10–100 mM H_2_O_2_), the mcm5S2U content decreased rapidly from 26% to 12% with a simultaneous increase in mcm5U (31–46%) and especially mcm5H2U (3–30%). This effect was due to the direct action of hydrogen peroxide on tRNA and means that the tRNA^Glu^ was probably directly damaged by the oxidizing agent because it damaged the yeast cells (the H_2_O_2_ concentration used was too high, above the “safe concentration”). The obtained results were shown in Figure 6 and Appendix A.

From the results obtained, we can conclude that R5S2U desulfuration occurs in yeast and the main reaction product is R5U, while the second desulfuration product, R5H2U, is present but maintained at a similarly low level in all the samples analyzed.

#### 2.4.2. Search for mcm5S2U Desulfuration Products in the tRNA Fraction of Cancer Cells

A fraction of small cellular RNAs (<200 nt) was isolated from total cellular RNA of the investigated cancer cells cultured without or in the presence of oxidizing reagents (H_2_O_2_, NaAsO_2_ or NaClO at IC_50_ concentrations, as seen in Table 2). This approach is described in the literature as one of the methods to obtain tRNA for research, as tRNAs represent the vast majority of low molecular weight RNAs [40]. Our tRNA isolation protocol has been described in detail in Materials and Methods. In brief, the cellular RNA fraction was fractionated on an Agilent SEC-3, 300Å HPLC column (150 mm × 7.8 mm) coupled with an FPLC AKTA purifier: Box-900, pH/C-900, UV-900, P-900, the Frac 920 system, and the low molecular weight RNA fraction (<200 nt) was separated from the large cellular RNAs, such as rRNAs and mRNAs. We chose this method to isolate the tRNA fraction containing not only one specific tRNA, as was in the case of yeast, but the mixture of all cellular tRNAs, because the amount of the specific tRNA isolated from the mammalian cells was much smaller compared to the tRNA from yeast cells and may not have been sufficient for further applications. The isolated tRNAs were subjected to nucleolytic hydrolysis (Benzonase, Phosphodiesterase 1) and dephosphorylation via alkaline phosphatase (CIAP) as described above. The resulting nucleoside mixture was analyzed by LC-MS/MS (ZenoTOF 7600 mass spectrometer coupled with ExionAC LC system, Sciex, Framingham, MA, USA) for the presence of mcm5S2U-tRNA desulfuration products. The LC-MS/MS experiments were repeated independently at least three times. The results of the analysis can be found in Figure 7 and Figure 8. Similar to the tRNA analysis of yeast, the percentages shown in the graphs indicate the proportion of the individual modified nucleosides in the nucleoside mixture. 100% was the sum of all 2-thiouridine derivatives tested.

First, we examined the content of the modified nucleosides (mcm5S2U, mcm5H2U, mcm5U, cm5S2U, cm5H2U, cm5U) in the tRNA from the cancer cells that were not exposed to exogenous oxidative stress, the so-called untreated cells. LC-MS/MS analysis revealed that in addition to the modified 2-thiouridines (mcm5S2U and cmS2U), which occurred in the highest quantities (29–53% and 33–57%, respectively), their desulfuration products (mcm5H2U, mcm5U, cm5U) were also present at the level dependent on the cell line examined, as seen in Figure 7A,B. The highest content of mcm5H2U was identified in the leukemic cells (MOLT-4 and K562, 11% and 6%, respectively) growing in suspension and in adherent melanoma cells (A375, 5%). The highest level of mcm5U was found in thr HeLa cells (37%), which distinguished the cells from the other cell lines where the average content of mcm5U was 4–8%. The cm5H2U modification was not identified in the tRNA from the cancer cells that were not subjected to oxidative treatment, which could be due to the low levels of this nucleoside, which were below the detection limit.

Next, we compared the effect of three oxidizing agents (H_2_O_2_, NaAsO_2,_ and NaClO, used individually in IC_50_ concentration, as seen in Table 2) on the formation of the 2-thiouridine desulfuration products in the low molecular weight (<200 nt) cellular RNA fraction. Figure 8 shows the presence of the desulfuration products in the three selected cell lines (HEK293, HeLa and A375). As expected, different oxidizing agents had different effects on the cells and at the same time the effect depended on the individual sensitivity of the cells tested. In the HEK293-unmodified cells, a significant decrease in the mcm5S2U content was observed after the application of oxidizing agents (from 43%, in untreated cells to the level of 24–20% in cells treated with H_2_O_2_, or NaAsO_2_ or NaClO, respectively). At the same time, the increase in mcm5S2U desulfuration products, mcm5H2U (from 2% in untreated cells to 6–14%) and mcm5U (from 5% in untreated cell to 11–12%), was observed. A significant increase in the mcm5H2U concentration (from 2% in untreated cells to 14–16%) was also observed in the Hela cells, especially after the cells were treated with H_2_O_2_ or NaAsO_2_. In the A375 cells, the significant decrease in the amount of mcm5S2U was observed (from 33% in untreated cells to 21%), leading to the increase in the amount of mcm5U (from 8% to 16%). However, the second desulfuration product remained at an approximately constant level (4–5%, similar as in the untreated cells). Interestingly, the cm5H2U modification, which was not present in the untreated cells, was detected in all the tested cells treated with H_2_O_2_, NaAsO_2_ and NaClO at the level of 3–5%.

We also performed a time-dependent experiment in which the cells were exposed to H_2_O_2_-induced oxidative stress (in IC_50_ concentration, Table 2) for 2, 4, or 6 h. LC-MS/MS analysis revealed that the highest decrease in the mcm5S2U level and the appearance of 2-thiouridine desulfuration products in the samples occurred during first 2 h and lasted up to 4 h of cell culture under oxidative stress conditions. Afterwards, the cells returned to the initial state, which can be explained by the activity of the cellular antioxidant system and the decomposition of the oxidizing agent. This effect was also dependent on the individual sensitivity of the cells of the respective cell line.

#### 2.4.3. Analysis of HEK293*ΔSOD2* and HEK293*ΔCat* tRNA Fraction Content

The experiment aimed to test the extent to which 2-thiouridine desulfuration occurs in cells with a partially damaged antioxidant system compared to unmodified HEK293 cells. For this purpose, cells were selected that showed no expression of the *Cat* or *SOD2* gene, Figure 1. The detailed protocol of selection is described in Section 2.1.1, Materials and Methods, and in Appendix A. Unmodified HEK293 cells, modified HEK293*ΔSOD2* cells (clones 1-1C6; 2-1 H12; 3-1D8), and HEK293*ΔCat* (clones 3-1E1; 2-2F7) cells obtained by the CRISPR/Cas9 gene editing protocol were exposed to oxidative stress during their growth (0–50 µM H_2_O_2_, IC_50_ = 14.5 µM H_2_O_2_ for HEK293; IC_50_ = 10 µM H_2_O_2_ for the HEK293 mutation variants, as seen in Table 2). The same procedure as described above was used to isolate the low molecular weight fraction of the cellular RNAs. The isolated tRNAs were subjected to nucleolytic hydrolysis (Benzonase, Phosphodiesterase 1) and dephosphorylation by alkaline phosphatase (CIAP). The resulting nucleoside mixture was analyzed by LC-MS/MS (ZenoTOF 7600 mass spectrometer coupled with ExionAC LC system, Sciex, Framingham, MA, USA) for the presence of mcm5S2U-tRNA desulfuration products. The synthetic nucleosides, mcm5S2U, mcm5H2U, mcm5U, cm5S2U, cm5H2U, and cm5U, served as nucleoside standards for LC-MS/MS analysis. The LC-MS/MS experiment was repeated twice independently. The results are shown in Figure 9.

The tRNA fraction isolated from the unmodified HEK293 cells was characterized by the high content of mcm5S2U (76–80%), accompanied by the low content of cm5S2U (4%). In the presence of H_2_O_2_, the level of mcm5S2U decreased slightly, while the gradual increase in mcm5H2U (0–9%), the putative desulfuration product, was observed. After the application of 50 µM H_2_O_2_, the content of mcm5H2U increased to 15%, which was probably due to the direct effect of peroxide on tRNA in the damaged cells.

The HEK293*ΔCat* deletion variants were as follows: the 3-1E1 clone, characterized by deletion of 14 bp in the CRISPR target sequence) and the 2-2F7 clone, without sequence changes in the target sequence (the changes probably occurred outside the sequenced target sequence), as seen in Appendix A, did not express the *Cat* gene, as seen in Figure 1. The tRNA fraction isolated from these cells contained the high content of mcm5S2U (89% in untreated cells) and the low level of cm5S2U (1–6%). After the application of 5 µM H_2_O_2_ to the culture, the level of mcm5S2U rapidly decreased to 60%, which was accompanied by the increase in the amount of mcm5H2U (from 7% to 24%) and mcm5U (from 2% to 12%). Further increasing the H_2_O_2_ concentration (10–50 µM) in the cell culture did not lead to an increase in the amount of 2-thouridine desulfuration products.

The HEK293*ΔSOD2* deletion variants were as follows: the 3-1D8 clone, characterized by deletion of 6 bp and numerous mutations outside the target sequence; the 1-1C6 clone, characterized by at least twelve point mutations and an insertion outside the target sequence; and a 2-1H12 clone, characterized by numerous mutations inside and outside the target sequence, which did not express the *SOD2* gene, as seen in Figure 1. The tRNA fraction isolated from these cells contained the high level of mcm5S2U (57–83% in the untreated cells) and the low level of cm5S2U (2–5%) depending on the clone. After the application of H_2_O_2_ to the culture, we observed the decrease in the amount of mcm5S2U (from 83 to 70%) with a simultaneous increase in the amount of mcm5H2U (from 7% to 16%) and mcm5U (from 7% to 11%) only in the case of the 1-1C6 clone. In the other two clones (3-1D8 and 2-1H12), the situation was reversed. We observed an increase in the amount of mcm5S2U and a decrease in the mcm5H2U, which is difficult to explain at this point.

In summary, the expected major changes in the course of the 2-thiouridine desulfuration process in HEK293 mutant variants with inactive *Cat* or *SOD2* genes were very subtle. A slight increase in the amount of mcm5S2U desulfuration products was observed, but this was dependent on the clone studied. Our observation can be explained by the fact that only individual *Cat* or *SOD2* genes were silenced in the cells. Knocking out all three genes in the one type of cell would most likely be fatal for the cells. In our case, the genes coding for the other enzymes of the antioxidant system remained active and the antioxidant enzymes reduced the effect of oxidative stress on the cells.

## 3. Materials and Methods

### 3.1. Chemical Synthesis of a R5-Substituted-2-Thiouridines and Derivatives

The nucleoside standards (mcm5S2U, mcm5H2U, mcm5U, cm5S2U, cm5H2U, cm5U, S2U, H2U, U) for LC-MS/MS analysis were synthesized at the Lodz University of Technology (TUL, Lodz, Poland) according to previously established and published protocols. The structures of the products were confirmed by NMR spectroscopy. 5-Methylcarboxymethyl-2-thiouridine (mcm5S2U) was synthesized using the “silyl method” of *N*-glycosidic bond formation between 5-ethylcarboxymethyl and 2-thiouracil and 1-*O*-acetyl-2,3,5-tri-*O*-benzoyl-*β*-D-ribofuranose. [41] followed by the transesterification of the ethyl ester and deprotection of the hydroxyl groups of the ribose moiety, which was performed in a 0.1 M solution of sodium methoxide in methanol. 5-Methylcarboxymethyluridine (mcm5U) was prepared by the malonylation of the appropriately protected 5-bromouridine, followed by a step of decarboxylation of the 5-malonyl moiety and deprotection of the nucleoside’s hydroxyl groups [40]. The desulfuration of the protected mcm5S2U was carried out with 3-chloroperoxybenzoic acid (mCPBA) to provide mcm5H2U [34]. The ester groups of the modified nucleosides were hydrolyzed under basic conditions (in 0.1 M KOH aqueous solution) to give appropriate acids: cm5S2U, cm5U, cm5H2U [34].

### 3.2. Protocols for Yeast

#### 3.2.1. Yeast Cell Culture

The following *Saccharomyces cerevisiae* yeast strains were used: the commercial INVSc1 strain (Thermo Fisher Scientific, Warsaw, Poland), the wild-type M3 strain, kindly provided by Prof. Michael Forte (Oregon Health & Science University, Portland, OR, USA) and the copper–zinc superoxide dismutase-depleted mutant M3 strain (M3*Δsod1*), kindly provided by Dr. Andonis Karachitos and Dr. Martyna Baranek (Adam Mickiewicz University, Poznan, Poland) [42,43]. The yeast cells were cultured overnight in a flask containing 50 mL of liquid YPD medium (1% yeast extract; 2% peptone; 2% glucose) in an incubator (30 °C) with vigorous shaking (200–250 rpm). The next day, fresh YPD medium (200 mL) was inoculated with 1% inoculum from the overnight culture and the yeast culture was continued in an Erlenmeyer flask for the next 3–4 h at 30 °C, until the optical density of the cells at λ = 600 nm (OD_600_) reached the value of 0.8–1.0. Then, the oxidizing reagent was added directly to the medium at different concentrations (H_2_O_2_: 0, 5, 10, 25, 50, 75, 100 mM; NaAsO_2_: 0, 10, 20, 40, 60, 80, 100 mM; NaClO: 0, 10, 20, 40, 60, 80, 100 mM), and the yeast growth continued under the same temperature and agitation conditions for 1 or 2 h. After the completion of the culture, the yeast cells were harvested by centrifugation (6000 rpm, 10 min at 4 °C), washed, counted, and subjected to further analysis or frozen at -80 °C until further use.

#### 3.2.2. Yeast Viability Assay

To determine the viability of the yeast cells, the spotting assay and the colony-forming unit (CFU) assay were performed according to the protocol of Kwolek-Mirek et al. [44]. The yeast grew in the liquid YPD medium under the conditions described above. When the culture reached the exponential growth phase (OD_600_ = 0.8–1.0) the yeast cells were exposed to oxidative stress. After 1 or 2 h of oxidative stress, the cells were centrifuged (6000 rpm, 10 min at 4 °C), washed in sterile PBS buffer, counted, and diluted to 10^7^, 10^6^, 10^5^, 10^4^, 10^3^ cells/mL. For the spotting assay, 5 µL of each suspension was spotted onto solid YPD medium (containing 1.5% agar) and incubated at 30 °C for 48 h, then the growth of yeast was observed and compared. For the colony-forming unit assay, 100 µL of the 10^3^ dilution was applied to solid YPD medium and incubated at 30 °C for 48 h. The colonies that grew on the plate were counted and compared to the other plates.

#### 3.2.3. Yeast Intracellular ROS Level

To determine the intracellular ROS content in the yeast cells exposed to oxidative stress, a H2DCF-DA (2′,7′-dichlorofluorescin diacetate, Sigma Aldrich, St. Luis, MO, USA) assay was performed. The yeast cells were grown in liquid YPD medium under the conditions described above and exposed to oxidative stress. After 1 or 2 h of incubation in oxidative stress, the culture was centrifuged (6000 rpm, 10 min at 4 °C), washed twice in PBS buffer (pH 7.4), diluted, counted, and the 1.5 × 10^7^ cells were incubated for 30 min in the dark with 10 µM H2DCF-DA reagent. Then, the cells were centrifuged, washed, and transferred to the black 96-well plate (Perkin Elmer, Warsaw, Poland). The fluorescence of the cells was measured directly using a Synergy HT plate reader (BIO-TEK, Altium International, Warsaw, Poland). The excitation and emission wavelengths for the fluorescein derivative were as follows: λ_ex_ = 485 nm and λ_em_ = 520 nm, respectively. The quantification of the data was performed with KC4 software v.3.4 (BIO-TEK, Altium International, Warsaw, Poland). The obtained results were given as the fluorescence intensity of the tested samples normalized to the fluorescence of not-treated control cells (CTR).

#### 3.2.4. γ-Toxin Preparation

Recombinant γ-toxin (from *Kluyveromyces lactis*) was overexpressed in the BL21(DE3) bacterial strain, (Thermo Fisher Scientific) containing the pET28-smt3 expression plasmid encoding the 10x His-Smt3-Ulp1-γ-toxin fusion protein, kindly provided by Prof. Steward Shuman (Memorial Sloan Kettering Cancer Institute, New York, NY, USA). The bacteria were cultured in 2 L of LB medium inoculated with 1% inoculum from the overnight culture and grown at 37 °C for approximately 4 h, until the exponential (Log) growth phase was reached: OD_600_ = 0.4–0.6. Then, the culture was cooled to 4 °C by incubation on ice, and 0.4 mM IPTG with 2% ethanol was added to induce γ-toxin overexpression. The culture was maintained at 17 °C with shaking (200 rpm) for the next 16 h. The culture was centrifuged (4000 rpm, 30 min, 4 °C), the pellet was washed with water to remove the residual medium, and it was frozen at -70 °C. For protein isolation, the bacterial pellet was suspended in buffer containing 100 mM Tris-HCl pH 8.8, 500 mM NaCl, 10% glycerol supplemented with Complete Protease Inhibitor, which was EDTA-free (Roche, Warsaw, Poland) and sonicated. After the completion of this step, the insoluble material was removed by centrifugation (17,000 rpm, 30 min, 4 °C). The supernatant was applied to the nickel affinity resin (Ni-NTA agarose, Thermo Fisher Scientific, Warsaw, Poland), equilibrated with the same buffer. The resin was washed with a buffer containing 50 mM Tris-HCl, pH 8.0, 250 mM NaCl, and 10% glycerol, and the pure protein was eluted with buffer containing 50 mM Tris-HCl, pH 8.0, 250 mM NaCl, 10% glycerol, and 500 mM imidazole. All the preparation steps were carried out at 4 °C. The protein was dialyzed to 50 mM Tris-HCl pH 8.0, and 100 mM NaCl was concentrated, frozen in liquid nitrogen, and stored at −80 °C until use. The results of the γ-toxin purification were placed in the Appendix A.

#### 3.2.5. γ-Toxin tRNA Digestion

The reaction was performed according to the protocol previously described by Lentini et al. [39]. Total cellular RNA samples (5 µg) isolated from yeast treated with H_2_O_2_ were incubated with γ-toxin (~14 µg /reaction) in 10 mM Tris-HCl pH 7.5, 10 mM MgCl_2_, 50 mM NaCl, and 1 mM DTT buffer for 15–30 min at 30 °C. The resulting reaction products were analyzed via a Northern blot using the tRNA^Glu^-specific DNA probe and RT-PCR with tRNA^Glu^-specific primers. The sequence of the probe and primers used can be found in the Appendix A.

#### 3.2.6. Northern Blot (NB) Analysis

The reaction mixtures after γ-toxin digestion (15 µL) were mixed with NB loading buffer (90% formamide with a mixture of tracking dyes: 0.01% bromophenol blue, 0.01% xylenocyanol), denatured for 2 min at 95 °C and loaded onto a 15% denaturing polyacrylamide gel for electrophoresis. After electrophoresis, the RNA was electro-transferred from the gel onto a Hybond-Ny+ membrane (Amersham Bioscience, Little Chalfont, UK). The membrane was placed in a tube and incubated with ULTRAhyb™ Ultrasensitive Hybridization Buffer (Thermo Fisher Scientific, Warsaw, Poland) at 42 °C for 1 h. Then, the ^32^P-labeled synthetic DNA probe complementary to yeast tRNA^Glu^ or control 25S rRNA was added and incubated overnight at 42 °C with slow rotation. The sequences of probes were as follows: a *S. cerevisiae* tRNA^Glu^ probe, 5′-ATAGCCGTTACACTATATCGGA-3′, and a positive control probe for *S. cerevisiae*, 25S rRNA 5′-GATTCTCACCCTCTATGACG-3′. The DNA oligonucleotides, which served as probes, were synthesized “in-house” at CMMS PAS using the routinely used phosphoramidite method on the H6 GeneWorld DNA/RNA synthesizer (K&A, Schaafhelm, Germany). The oligonucleotides were radiolabeled at the 5′ end with [γ-^32^P]-ATP (Hartman Analytic, Brunswick, Germany) in a reaction catalyzed by T4 polynucleotide kinase (New England Biolabs, Ipswich, MA, USA), according to the manufacturer’s protocol. After overnight incubation membrane with a appropriate probe, the unbound probes were washed out twice for 10 min (SSPE 2×, 0.1% SDS buffer); the membranes were air-dried, placed in a cassette with X-ray film, and exposed to radiation for several hours at room temperature in the dark.

#### 3.2.7. qRT-PCR Analysis

Real-time qRT-PCR reactions were performed with total cellular RNA from *S. cerevisiae* that was (i) not treated with γ-toxin, [γ-toxin (-)], or (ii) after γ-toxin hydrolysis [γ-toxin (+)]. The RNA (1 µg), the tRNA^Glu^-specific primers, and the components of the LightCycler^®^ RNA Amplification SYBR Green I kit (Roche, Warsaw, Poland) were mixed according to the manufacturer’s protocol. The reaction was performed on the Light Cycler instrument (Roche, Warsaw, Poland) in capillaries in 10 μL solution. The primers used in the experiment were specific for *S. cerevisiae* (sc) tRNA^Glu^, sc-tRNA^Glu^ Fw: 5′-TCCGATATAGTGTAACGGCTAT-3′; sc-tRNA^Glu^ Rv: 5′-CTCCGATACGGGGAGTCG-3′, and specific for the control, *S. cerevisiae* 25S rRNA: 25S rRNA Fw 5′-GAAATCTGGTACCTTCGGTG-3′ and 25S rRNA Rv 5′-GATTCTCACCCTCTATGACG-3′. The following program was used for the reaction: reverse transcription at 55 °C for 10 min and reverse transcriptase denaturation at 95 °C for 30 s; PCR (45 cycles) at 95 °C for 10 s, 60 °C for 10 s, and 72 °C for 10 s; and Tm of the PCR products analysis according to the Light Cycler instructions.

#### 3.2.8. Yeast Total RNA Preparation and Pull-Down of tRNA^Glu^

The yeast samples (e.g., 2 g) were suspended in 0.5 mL sterile water and then in 5 mL TRI Reagent (Thermo Fisher Scientific, Warsaw, Poland) with 2 g glass beads (0.5 mm) (Sigma Aldrich, St. Luis, MO, USA). The yeast suspension was vortexed vigorously to efficiently perform cell lysis. Total cellular RNA was prepared according to the TRI Reagent protocol via extraction with chloroform and precipitation of the RNA with isopropanol. The resulting pellet was washed with 70% ethanol and air-dried. The RNA concentration was determined spectrophotometrically by measuring the absorbance at 260 nm.

The tRNA^Glu^ was isolated from the total cellular RNA via the pull-down method with a specific biotinylated probe complementary to the anticodon loop domain (5′-biotin-CCGGTCTCCACGGTGAAAGCGTGATGTGATAG-3′) bound to the streptavidin–agarose resin (Thermo Fisher Scientific, Warsaw, Poland). Approximately 500 μL of the streptavidin–agarose slurry was added to the sterile Eppendorf tube, washed three times with (i) RNase-free deionized water, then three times with (ii) 100 mM sodium phosphate buffer pH 7.4 containing 125 mM NaCl, and then incubated with 15 nmol of the biotinylated DNA probe for 1 h at room temperature with vigorous shaking (~2000 rpm). After incubation, the agarose beads were washed several times (at least six times) with 100 mM sodium phosphate buffer/125 mM NaCl to remove the unbound oligonucleotide. The washing step was monitored spectrophotometrically (Nanodrop, BIO-TEK, Altium International, Warsaw, Poland). Approximately 500 mg of the total RNA solution was added to the agarose beads, incubated at 80 °C for 5 min, and cooled and incubated overnight with vigorous shaking (~2000 rpm) at 4 °C. The next day, the resin was washed at least six times with 100 mM sodium phosphate buffer, pH 7.4, until the absorbance of the wash solution reached 0. The bound tRNA^Glu^ was eluted via the addition of 200 μL deionized water, incubation at 80 °C for 2 min, and centrifugation (3000 rpm, 3 min). The elution step was repeated at least six times. The solution with eluted tRNA^Glu^ was lyophilized using the vacuum SpeedVac concentrator (Savant, Thermo Fisher Scientific, Warsaw, Poland). The amount of isolated tRNA^Glu^ was determined spectrophotometrically at λ = 260 nm. The samples were stored at −80 °C until further use.

### 3.3. Protocols for Human Cells

#### 3.3.1. Human Cell Culture

All the human cell lines used in the studies were from the European Collection of Authenticated Cell Cultures (ECACC), distributed by Sigma Aldrich. The human cells HeLa, K562, MOLT-4, A375, and A549 were cultured in RPMI-1640 medium (Gibco, Thermo Fisher Scientific, Warsaw, Poland), supplemented with 10% fetal bovine serum (FBS) (Gibco, Thermo Fisher Scientific, Warsaw, Poland), and the antibiotics penicillin–-streptomycin (Gibco, Thermo Fisher Scientific, Warsaw, Poland), HEK293, and modified HEK293 (obtained “in-house”, CMMS PAS, as described below), A431, U-87 MG, and MCF-7 (cultured in DMEM medium (Gibco, Thermo Fisher Scientific, Warsaw, Poland), supplemented with 10% FBS (Gibco, Thermo Fisher Scientific, Warsaw, Poland), and the antibiotics penicillin and streptomycin (Gibco, Thermo Fisher Scientific, Warsaw, Poland). The complete list of the cell lines, their origin, and their culture conditions can be found in the Appendix A.

#### 3.3.2. Cytotoxicity Assay

The cytotoxic effect of the oxidizing agents (H_2_O_2_, NaAsO_2_ or NaClO) on the viability of the tested cells was determined using an MTT assay. The cell suspension of the 7 × 10^3^ cells per well was seeded in a 96-well plate in 200 µL of an appropriate medium, as seen in Appendix A. After 24 h of incubation at 37 °C in the 5% CO_2_ atmosphere, the culture medium was replaced with fresh medium containing the respective reagent at the following concentrations: 0, 5, 10, 25, 50, 100, and 200 µM (H_2_O_2_, NaAsO_2_), or 0, 1, 5, 10, 20, 50, and 100 µM (NaClO). The cell culture under oxidative stress conditions was continued for the next 1 or 2 h. The untreated cells were used as positive control (100% viability) and the medium without cells was used as the background. After the incubation, 25 µL of MTT (3-(4,5-dimethylthiazol-2-yl)-2,5-diphenyltetrazolium bromide, Sigma Aldrich, St. Luis, MO, USA) solution in PBS (5 mg/mL) was added to each well and the cells were incubated at 37 °C for another 2 h. Finally, 95 μL of hot lysis buffer (20% SDS, 50% aqueous dimethylformamide, pH 4.5) was added to each well and the plates were incubated overnight at 37 °C. The absorbance of the lysed cells was measured at 570 nm and 630 nm using the FLUOstar Omega plate reader (BioGenet, Jozefow, Poland) and ΔAbs = Abs_570_ − Abs_630_ was calculated. The results were presented as a dependence of the percentage of live cells on the concentration on the oxidizing agent and the IC_50_ coefficient was determined for each cell line.

#### 3.3.3. Intracellular ROS-Level Determination

The intracellular ROS content was determined using the DCFDA (2′,7′-dichlorofluorescin diacetate) Cellular ROS Detection Assay Kit (Abcam, Cambridge, UK) according to the manufacturer’s protocol after the cells had been exposed to the oxidizing mixture (H_2_O_2_, NaAsO_2,_ or NaClO).

Adherent cells (HeLa, A375, A549, A431, U-87 MG, MCF-7, HEK293) were grown in RPMI-1640 or DMEM medium supplemented with 10% FBS and antibiotics as required. The cells were harvested and the suspension of 25 × 10^3^ cells/well was seeded into a clear bottom black 96-well microplate (PerkinElmer, Warsaw, Poland). The cells were allowed to grow overnight, then washed with PBS buffer and incubated with 25 µM DCFDA solution at 37 °C with 5% CO_2,_ in the dark. After 45 min of incubation, the cells were washed in PBS buffer and treated with oxidants (0, 5, 10, 25, 50, 100, 200 µM H_2_O_2_; 0, 5, 10, 25, 50, 100, 200 µM NaAsO_2_; or 0, 1, 5, 10, 20, 50, 100 µM NaClO) in the 100 µL/well RPMI-1640 or DMEM colorless medium containing 10% FBS and antibiotics. The fluorescence of the cells was measured after 0, 1, 2 and 4 h of incubation using the FLUOstar Omega plate reader (BioGenet, Jozefow, Poland) at E_x_/E_m_ = 485/520 nm in endpoint mode in the presence of oxidizing compounds in the medium.

Suspension cells (K562 and MOLT-4) were grown in RPMI-1640 medium supplemented with 10% FBS and antibiotics. Then, the cells were harvested by centrifugation, washed in PBS buffer and stained with a 20 µM DCFDA solution of 1 × 10^6^ cells/mL, at 37 °C, in the dark. After incubation for 30 min, the cells were washed in PBS, suspended in colorless RPMI-1640 medium containing 10% FBS and antibiotics to 1 × 10^5^ per well, seeded in clear bottom black 96-well microplates (PerkinElmer, Warsaw, Poland) and exposed to the appropriate oxidant used as described above. The fluorescence of the cells was measured after 0, 1, 2 and 4 h, using the FLUOstar Omega plate reader (BioGenet, Jozefow, Poland) at E_x_/E_m_ = 485/520 nm in the endpoint mode in the presence of the oxidizing compounds in the medium.

#### 3.3.4. Isolation of Total Cellular RNA and the Total Cellular tRNA Fraction

The human cells (3 × 10^6^) were grown in the culture flask (25 cm^2^) in 5 mL of a suitable medium containing 10% FBS and antibiotics for 24 h. Then, the cells were exposed to the oxidizing agent used at a concentration determined by the IC_50_ for each cell line, as described above. After 2 h of incubation of the cells in the oxidizing environment, the cells were washed with PBS buffer, and harvested and lysed in 1 mL of TRI Reagent (Thermo Fisher Sci., Warsaw, Poland). The total cellular RNA was prepared according to the TRI Reagent protocol. The resulting total RNA preparation was stored at −80 °C until use. The RNA concentration was determined spectrophotometrically by measuring the absorbance at 260 nm.

Further steps to isolate the total cellular tRNA fraction followed the procedure described by Su et al. [36]. In brief, the total RNA samples were denatured by incubation at 80 °C for 2 min and the total cellular tRNA fraction was isolated using an Agilent SEC-3, with a 300Å HPLC column and a 150 mm length × 7.8 mm inner diameter (Perlan Technologies, Warsaw, Poland) on an FPLC AKTA purifier: Box-900, pH/C-900, UV-900, P-900, and a Frac 920 system (GE, Warsaw, Poland) in the aqueous 100 mM ammonium acetate phase, with a flow rate of 0.5 mL/min. The resulting tRNA fraction was precipitated in ice-cold ethanol (2.5 volumes) in the presence of 3 M sodium acetate, with a pH of 5.2, (0.1 volumes), and stored at −80 °C until use.

#### 3.3.5. tRNA Hydrolysis to Nucleosides

The tRNA hydrolysis was performed according to the described protocols [36,45]. In brief, the tRNA sample (~10 µg) was hydrolyzed with a combination of two nucleases; Benzonase (endonuclease from *Serratia marcescens*, Sigma-Aldrich, St. Luis, MO, USA), with 20 units per reaction, in 50 mM Tris-HCl with a pH of 8.0, of 1 mM MgCl_2_, for 4 h at 37 °C; Phosphodiesterase I from *Crotalus adamanteus* venom (Sigma-Aldrich, St. Luis, MO, USA), with 0.8 units per reaction, in 50 mM Tris-HCl, of 20 mM MgCl_2_, for 16 h at 37 °C; and, finally, alkaline phosphatase (EURX, Gdansk, Poland), with 10 units per reaction, in manufacturer’s buffer (1 M diethananolamine, 10 mM, p-nitrophenylophosphate, 0.25 mM MgCl_2_, pH 9.8) for 1 h at 37 °C. The sample after hydrolysis was filtered with a 10,000 MW cut-off spin filter (Amicon, Merck Millipore, Poland) and dried in a vacuum centrifuge (Savant, Thermo Fisher Scientific, Warsaw, Poland).

### 3.4. CRISPR/Cas9 Genome Editing Experiments

#### 3.4.1. Transfection

HEK293 cells were seeded in a 6-well plate, at 3 × 10^5^ cells per well, in a high glucose DMEM medium supplemented with 10% FBS and antibiotics (Gibco, Thermo Fisher Scientific, Warsaw, Poland). After 24 h incubation at 37 °C, 5% CO_2_, a cell confluence of about 60% was reached. The transfection of the sgRNA-Cas9 complex was performed according to the instruction of Lipofectamine “CRISPRMAX” (Thermo Fisher Scientific, Warsaw, Poland). Two mixtures were prepared for one well of the 6-well plate. Tube 1 contained 6.25 µg of TrueCut Cas9 v2 (Thermo Fisher Scientific, Warsaw, Poland), which was mixed with 1.2 µg of sgRNA (Horizon Discovery, ABO, Gdansk, Poland) and Cas9 Plus Reagent (12.5 µL) in 125 µL of OPTI MEM I medium (Gibco, Thermo Fisher Scientific, Warsaw, Poland). Tube 2 contained Lipofectamine “CRISPRMAX” (7.5 µL), which was diluted in OPTI MEM I medium (125 µL). The solution from Tube 1 was immediately added to Tube 2; the mixture was incubated for 10 min at room temperature and the formed complex was added to the cells. The HEK293 cells were incubated at 37 °C in 5% CO_2_ for the next 3 days. After incubation, the cells from each well were counted, diluted to a concentration of approximately 1 cell per 100 µL of medium, and aliquoted into 96-well plates (three 96-well plates per one sgRNA applied, for a total of 9 plates for one gene) to obtain single clones. The rest of the cells were washed with PBS and stored at −70 °C for further testing.

#### 3.4.2. Western Blot

The proteins contained in the 20–40 µg cell lysate were separated by electrophoresis in 10–15% SDS-PAGE and electro-transferred from the gel to Immobilon-P PVDF membrane (Sigma Aldrich, St. Luis, MO, USA) via rapid transfer buffer (VWR International, Gdansk, Poland). The PVDF membrane was blocked via incubation in 5% non-fat milk in TBS buffer (20 mM Tris-HCl pH 7.4, 0.9% NaCl) and then incubated overnight at 4 °C with one of the following primary antibodies: rabbit anti-Cat (1:1000), mouse anti-SOD1 (1:1000), rabbit anti-SOD2 (1:1000), or reference mouse anti-GAPDH (1:1000) (Cell Signaling, Danvers, MA, USA). The next day, the membranes were washed 3 times for 10 min each in TBST buffer (TBS with 0.1% Tween 20) and incubated with secondary antibodies against rabbit or mouse conjugated with alkaline phosphatase, with a dilution of 1:4000 (Sigma-Aldrich, St. Luis, MO, USA), for 2 h at room temperature. The membranes were then washed 3 times for 10 min each in TBST buffer and incubated for 5 min at room temperature with a chemiluminescent substrate for alkaline phosphatase (DuoLuX^®^ Chemiluminescent and Fluorescent Substrate, Alkaline Phosphatase, Vector Lab., Biokom, Janki, Poland). The protein bands were visualized with the gel visualization system (Uvitec, Cambridge, UK).

#### 3.4.3. T7 Endonuclease I-Based Mutation Detection

Genomic DNA was isolated from modified HEK293 cells using the Genomic Mini AX Tissue Spin kit (A&A Biotechnology, Gdansk, Poland) according to the manufacturer’s protocol. The PCR primers were designed for the detection of mutations in the human genes *Cat*, *SOD1*, or *SOD2*, caused by the CRISPR/Cas9 genome editing system. The primer sequences for the three target genes are listed in Appendix A. All DNA oligonucleotides were synthesized “in-house” (CMMS PAS, Lodz, Poland). The PCR reaction was performed under the conditions described in the T7 endonuclease 1-based mutation detection protocol in the EnGen Mutation Detection kit (NewEngland Biolabs, Ipswich, MA, USA). The PCR mixture contained the following: 200 ng of genomic DNA, 12.5 µL of Q5 Hot Start High-Fidelity (2×) master mix, 0.5 µM of appropriate forward and reverse primers, and nuclease-free water up to 25 µL. Thermocycling was performed under the following conditions: (1) one cycle of initial denaturation at 98 °C for 30 s; (2) 35 cycles of denaturation at 98 °C for 5 s, with annealing at 62 °C for 10 s, and extension at 72 °C for 20 s; and (3) the final extension at 72 °C for 2 min. 

A small amount of PCR product (e.g., 5 µL) was analyzed on agarose gel to check the amplification of a single product of correct size compared to the DNA ladder size.

In terms of heteroduplex formation, the mixture of 5 µL of PCR product with 2 µL of NEB buffer2 (10×) and 12 µL of ddH2O was denatured at 95 °C for 5 min and then cooled to room temperature for 20 min. Then, 1 µL of EnGen T7 endonuclease was added and the heteroduplex digestion reaction was incubated at 37 °C for 15 min. After digestion, 1 µL of Proteinase K was added to the mixture and incubated at 37 °C for 5 min to inactivate the T7 endonuclease-1. The products of heteroduplex digestion were analyzed on the agarose gel.

### 3.5. Analysis of tRNA-Derived Nucleosides by LC-MS/MS

All the described LC-MS/MS analyses were performed in the LabExperts laboratory (Lodz, Poland). The LC-MS/MS analysis of the nucleoside mixture derived from yeast tRNA was carried out on a QTRAP 6500+ mass spectrometer (Sciex, Framingham, MA, USA), coupled with the Exion LC System (Sciex, Framingham, MA, USA). The LC-MS/MS analysis of the nucleoside mixture derived from human tRNA was carried out on a ZenoTOF 7600 mass spectrometer (Sciex, Framingham, MA, USA), coupled with the ExionAC LC System (Sciex, Framingham, MA, USA). Chromatographic separation was conducted on an InfinityLab Poroshell 120 PFP, 2.1 × 100 mm, 2.7 µm, with a narrow bore LC column (Agilent, Santa Clara, CA, USA), thermostated at 50 °C. The major chromatography parameters were as follows: injection volume, 30 μL; constant flow, 0.5 mL/min; the mobile phase combining solvent A: 0.1% formic acid in water (LC-MS grade) and solvent B: 0.1% formic acid in acetonitrile (ACN) (LC-MS grade), with a total analysis time of 7 min. The gradient started with 0.5 min of pre-injection conditioning, followed by gradient separation starting from 2% B, increasing to 5% B after 1 min., followed by a further increase to 60% B after 4 min, left at 60% B for 5 min. The initial conditions were restored from 5.1 to 7 min. of the run. A summary of the applied gradient conditions is provided in Appendix A. The development of the method included the optimization of the ion source parameters, the selection of fragment ions, and the optimization of MRMhr parameter scanning. An electrospray ion source (ESI) was used. The detailed parameters are presented in the Appendix A. The MS/MS detection was made in positive ionization-scheduled multiple reaction-monitoring (sMRM) mode. The optimized ESI ions source parameters were as follows: CUR: 25; IS: 5000 V; TEMP: 400 °C; GS1: 50; and GS2: 40. The quantitation of the monitored compounds was performed on the basis of the standard curves prepared for synthetic nucleoside standards within a range of concentrations from 0.01 nM to 1 µM.

## 4. Conclusions

Aerobic organisms, including humans, use oxygen to produce energy in the mitochondrial respiratory chain, in which the oxygen molecule undergoes a four-electron reduction, and the energy released during this process is used to generate ATP. The electron flow through the respiratory chain is not completely dense; some electrons “escape” and reduce the oxygen via a one-electron process. The result of this reduction is a superoxide anion radical (O_2_^•−^), which is converted into hydrogen peroxide (H_2_O_2_) and other oxygen species. It has been estimated that about 1–4% of the oxygen consumed by the mitochondria undergoes one-electron reduction and that this is the main source of the O_2_^•−^ radical in aerobic cells [46,47,48]. Reactive oxygen species (ROS) can also be generated in the body by external physical factors such as ionizing radiation, ultraviolet radiation, and ultrasound, as well as by some intracellular enzymatic reactions catalyzed by oxidases, reductases, dehydrogenases, etc. Aerobic cells have developed a defense system against the destructive effect of ROS, consisting of low molecular antioxidants, antioxidants, and repair enzymes. In addition to the undoubtedly positive functions that ROS fulfill in the body, e.g., as components of signal transduction pathways, in cell cycle progression, in differentiation, in cell development and apoptosis, when the generation of ROS exceeds their degradation via antioxidant enzymes, a state of oxidative stress arises in the cell. Excessive ROS concentrations cause oxidative damage to proteins, nucleic acids, polysaccharides, and lipids, finally leading to cell dysfunction and cell death. ROS react non-specifically with cellular components, modifying and damaging them. Oxidatively damaged biomolecules impair cellular functions and contribute to the pathology of a variety of disorders.

One of the molecules that fulfills the irreplaceable function in the cell and is susceptible to damage by ROS is transfer nucleic acid. Understanding the mechanisms of damage to modified nucleosides that occur in natural cellular tRNAs, and the consequences of this damage, is essential for expanding knowledge of the regulation of gene expression at the codon–anticodon interaction level. We have focused our interest on eukaryotic (yeast or human) tRNA containing 5-substituted-2-thiouridine (R5S2U) in the wobble position of the anticodon. Under the conditions of external or internal oxidative stress caused by the addition of the oxidizing reagent to the medium during culture, R5S2U was desulfurized. Using the highly sensitive and selective technique LC-MS/MS, we analyzed the nucleosides derived from the cellular tRNA^Glu^ or the low molecularweight RNA fraction (<200-nt) and detected, not only the modifications mcm5S2U or cm5S2U, but also the desulfuration products mcm5H2U, cm5H2U, mcm5U, and cm5U, confirming our hypothesis that 5-substituted S2U-containing tRNAs can be damaged in growing cells exposed to oxidative stress. The amount and type of R5S2U desulfuration products depended on the type of cells studied (yeast or human), the type of oxidation reagent (H_2_O_2_, NaAsO_2_ or NaClO), their concentration in the culture medium, and the exposure time. The observed changes in the cellular content of R5S2U and the formation of its desulfuration products, R5H2U and R5U, were quite subtle and differed markedly from the reactions performed in vitro, in the absence of cells, where the S2U desulfuration efficiency of 100% was easily achieved. This effect could be explained by the activity of the antioxidant system in the cells. The disruption of this system, e.g., by inhibiting the expression of genes coding for antioxidant enzymes, led to the significant sensitization of the cells to oxidative stress and to the slightly stronger effect of R5S2U-tRNA desulfuration. The process of R5S2U-tRNA desulfuration triggered by prolonged oxidative stress in living cells can impair the function of transfer RNAs and alter the translation of genetic information.

## Data Availability

All the relevant data are within the paper and Appendix A.

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
