# Peer review of "Studies on the Oxidative Damage of the Wobble 5-Methylcarboxymethyl-2-Thiouridine in the tRNA of Eukaryotic Cells with Disturbed Homeostasis of the Antioxidant System"

_ijms, 2024, doi:10.3390/ijms252212336_

Round 1
Reviewer 1 Report
Comments and Suggestions for Authors
The paper of Sierant et al "Studies on oxidative damage..." studied the desulfuration of 5-substituted 2-thiouridines in tRNAs caused by oxidative stress in yerast and in mammalian cells. The paper is interesting, present a lot of experimental data and it is very informative. There are only some minor questions related to a better description of Material and Methods used .
In details:
Lines 167-173, Please give a reference or the sources of the yeast strains used. The strains are also indicated in Table S1, but no informations or references are given !! Also the cited reference 37 and 38 seem not pertinent here..
Lines 180- 184 No references or information on the origin of the used cell lines is given, also in Table S2 these informations are lacking..
Lines 194-204 The paragraph could be withdrawn since is a description of the well known CRISPR-Cas9 method..
Lines 214 Edit-R sgRNAs obtained from??, True Cut Cas9 obtainef from ?, CRISPMAX reagents obtained from?? These information should be added here or in the Mat and Methods section.
The sequence of oligos used are in Table S3, but again no information are given about the origin (were synthetised in the lab or purchased by a Company??)
Fig3 What is ROS % ? is the fraction of positive cells?? please explain since the assay is fluorimetric and therefore a level of fluorescence is measured..
Line 593 please give a refrence for the "published protocols " used. The reference cited here (41) is not pertinent..
Lines 645-47 what is the source of BL21(DE3) E.coli strain and of the plasmid pET28-smt3 used here..(or give a reference..)
Lines 681 32P-labeled DNA probes , give the method used for labelling and for the synthesis of the used probes..
Line 784 The human cells ?? which cells ?? Hek 293 ??
Line 816-17 CRISPRMAX obtained from??
Author Response
Response for Reviewer 1
Thank you for your positive and very detailed evaluation of our manuscript. We appreciate for drawing our attention to the fact that some important information about the origin of the cells and other materials we used in our studies is missing. This information has been added to the relevant protocols in the Materials and Methods or another part of the manuscript. All changes in the revised manuscript are highlighted in red. Below you will find our responses to individual questions and suggestions.
Lines 167-173, Please give a reference or the sources of the yeast strains used. The strains are also indicated in Table S1, but no information or references are given !! Also the cited reference 37 and 38 seem not pertinent here
The text “The following Saccharomyces cerevisiae yeast strains were used: the commercial INVSc1 strain (Thermo Fisher Scientific), the wild type M3 strain, kindly provided by Prof. Michael Forte (Oregon Health & Science University, US) and the copper-zinc superoxide dismutase depleted M3 mutant strain (M3Δsod1), kindly provided by Dr Andonis Karachitos and Dr Martyna Baranek (Adam Mickiewicz University, Poznan, Poland)” was added in section 3.2.1. “Yeast cell culture”, lines 605-609 in the Materials and Methods.
The cited references 37 and 38 were changed.
Lines 180- 184 No references or information on the origin of the used cell lines is given, also in Table S2 these information are lacking
The information on the origin of the cell lines used was added in Materials and Methods, section 3.3.1. “Human cells culture”, lines 733-734: “All human cell lines used in the studies are from the European Collection of Authenticated Cell Cultures (ECACC), distributed by Sigma Aldrich” and in line 738: “and modified HEK293 (obtained “in-house”, CMMS PAS, as described below)”.
Lines 194-204 The paragraph could be withdrawn since is a description of the well-known CRISPR-Cas9 method
The paragraph (lines 195-204) describing the CRISPR/Cas9 method was removed
Lines 214 Edit-R sgRNAs obtained from??, True Cut Cas9 obtained from ?, CRISPMAX reagents obtained from?? These information should be added here or in the Mat and Methods section
The information on the origin of the reagents used has been added in Materials and Methods, section 3.4.1. “Transfection” (lines 815-830), highlighted in red. “HEK293 cells were seeded in 6-well plate, 3x 105 cells per well, in high glucose DMEM medium supplemented with 10% FBS and antibiotics (Gibco, Thermo Fisher Scientific). After 24 hours incubation at 37 °C, 5% CO2, when a cell confluence of about 60% was reached. transfection of the sgRNA-Cas9 complex was performed according to the instruction of Lipofectamine “CRISPRMAX” (Thermo Fisher Scientific). Two mixtures were prepared for one well of the 6-well plate. Tube 1: 6.25 µg of TrueCut Cas9 v2 (Thermo Fisher Scientific) was mixed with 1.2 µg of sgRNA (Horizon Discovery) and Cas9 Plus Reagent (12.5 µL) in 125 µL of OPTI MEM I medium (Gibco, Thermo Fisher Scientific)”.
The sequence of oligos used are in Table S3, but again no information are given about the origin (were synthetized in the lab or purchased by a Company??)
The information on the origin of oligonucleotides has been added in Materials and Methods, section 3.4.4. “T7 Endonuclease I based mutation detection” (lines 851-852) “All DNA oligonucleotides were synthesized “in-house” (CMMS PAS)”
Fig3 What is ROS % ? is the fraction of positive cells?? please explain since the assay is fluorimetric and therefore a level of fluorescence is measured
The Figure 3 was changed and in the capture the following text was added “The obtained results are given as the fluorescence intensity of the tested samples normalized to fluorescence of not treated control cells, CTR.” The explanation can be found in the description of Figure 3 and in Materials and Methods, section 3.2.3 “Yeast intracellular ROS level”
Line 593 please give a refrence for the "published protocols " used. The reference cited here (41) is not pertinent
The references were corrected ([42], [43] and [34])
Lines 645-47 what is the source of BL21(DE3) E.coli strain and of the plasmid pET28-smt3 used here..(or give a reference..)
The information about the origin of the BL21(DE3) bacterial expression strain and the pET28-smt3-g-toxin expression plasmid has been added: “BL21(DE3) (Thermo Fisher Scientific)………..plasmid… kindly provided by Prof. Steward Shuman (Memorial Sloan-Kettering Cancer Institute, New York, US)”.
Lines 681 32P-labeled DNA probes , give the method used for labelling and for the synthesis of the used probes
The following text was added in section 3.2.6. “Northern blot (NB) analysis”, in the Materials and Methods. “The DNA oligonucleotides, which served as probes were synthesized “in-house” at CMMS PAS using the routinely used phosphoramidite method on the H6 GeneWorld DNA/RNA synthesizer (K&A). The oligonucleotides were radiolabeled at the 5' end with [g-32P]-ATP (Hartman Analytic) in a reaction catalyzed by T4 polynucleotide kinase (New England Biolabs), according to the manufacturer's protocol”.
Line 784 The human cells ?? which cells ?? Hek 293 ??
This section of Materials and Methods describes the general protocol for all cell lines mentioned in 3.3.1 “Human cells culture” and Table S2 (HeLa, K562, MOLT-4, A375, A549, A431, U-87 MG, MCF-7, HEK293, HEK293DSOD2 and HEK293DCat).
Line 816-17 CRISPRMAX obtained from??
The information has been added “Lipofectamine “CRISPRMAX” (Thermo Fisher Scientific)”.

Reviewer 2 Report
Comments and Suggestions for Authors
The authors investigated the effect of oxidative stress on mcm5S2U in tRNA by quantitative LC-MS/MS-MRMhr analysis. They monitored decrease mcm5S2U, meantime the increace of desulfuration products mcm5H2U and mcm5U, under the treatment of H2O2, NaAsO2, NaClO. These results suggest that oxidative stress can impair the function of 2-thiouridine-containing tRNAs.
Here I have some concerns/suggestions below.
1. Knocking out the Cat or SOD2 gene is a straightforward approach to confirming the effect of oxidative stress on 2-thiouridine. However, the results provided do not make this effect clear.
2. Can the author clarify the response the response of cm5S2U to oxidative stress? For example, cm5S2U does mostly not decrease under oxidative treatment.
3. A few spelling errors need correction.
Author Response
Response for Reviewer 2
Thank you for the positive review of our manuscript. All changes in the revised manuscript are highlighted in red. Below you will find our responses to your individual questions and suggestions.
Knocking out the Cat or SOD2 gene is a straightforward approach to confirming the effect of oxidative stress on 2-thiouridine. However, the results provided do not make this effect clear.
Indeed, the changes in the level of 5-substituted 2-thiouridine desulfuration products after silencing the expression of the Cat and SOD2 genes are not as significant as we expected.
The changes in the content of 5-substituted-2-thiouridine and its desulfuration products formed under the influence of oxidative stress in tRNAs of the live cells are not as obvious and clear as in the in vitro reaction in the test tube, where the efficiency of S2U desulfuration reached 100%. The reaction in living cells, both in yeast (results in Figure 6 and Figure S12) and in human cells (Figure 9), is rather subtle. We observed only small but visible changes in the level of mcm5S2U, mcm5H2U or mcm5U. These results can be explained by the action of the cellular antioxidant system, which undoubtedly includes catalase and superoxide dismutase. Disruption of elements of the antioxidant system by blocking the gene expression of SOD2 or Cat resulted in a subtle sensitization of the modified cells to hydrogen peroxide compared to the unmodified cells (which is also reflected in the reduction of IC50, from 14.5 µM to 10 µM). In the unmodified HEK293 cells exposed to H2O2 (0-10 µM), the level of mcm5S2U decreased very slightly (80-76%), and a gradual increase in mcm5H2U (0-9%) was observed. After the application of 50 µM H2O2, the level of mcm5H2U increased to 15%, which was probably due to the effect of direct oxidation of tRNA in the damaged cells. In the case of the HEK293DCat deletion variant (e.g. in the 2-2F7 clone), the level of mcm5S2U rapidly decreased from 89% to 60% when 5 µM H2O2 was applied to the culture, which was accompanied by an increase in the amount of mcm5H2U (from 7% to 24%) and mcm5U (from 2% to 12%). A further increase in H2O2 concentration (up to 50 µM) in the cell culture did not lead to an increase in the amount of 2-thiouridine desulfuration products. In HEK293DSOD2 mutation variants after application of H2O2 to the culture, we observed the decrease in the amount of mcm5S2U (from 83 to 70%) with a simultaneous increase in the amount of mcm5H2U (from 7% to 16%) and mcm5U (from 7% to 11%) only in the case of 1-1C6 clone.
- Can the author clarify the response of cm5S2U to oxidative stress? For example, cm5S2U does mostly not decrease under oxidative treatment.
The cm5S2U and cm5U modifications were identified in all cells tested, both in yeast and cancer cells as well as in HEK293 mutants. In the M3 strain of S. cerevisiae, the nucleosides cm5S2U and cm5U remained at the constant level despite the use of H2O2 at a concentration of 0-25 mM. Only the use of a high concentration of H2O2 (100 mM), which directly damaged the cells and oxidized the tRNA, led to a rapid decrease in cm5S2U and the formation of larger amounts of cm5H2U. In M3Dsod1 mutants, which are more sensitive to hydrogen peroxide, this process began when 10 mM H2O2 was used. In cancer cells and in normal model cells, the level of modified nucleosides also remained at a constant level and was dependent on the line tested and the oxidizing agent used. After application of the oxidizing agent, the desulfuration product cm5H2U also appeared, which was not present in untreated cells. According to the information collected in the Modomics RNA modifications database, cm5S2U and cm5U are substrates for the respective methyltransferases for the formation of mcm5S2U or mcm5U (https://genesilico.pl/modomics/protein/274/), therefore we believe that the applied oxidative stress possibly inhibited the activity of these enzymes (partial inactivation?), resulting in the accumulation of these modified nucleosides in the respective tRNAs.
- A few spelling errors need correction.
The manuscript was rechecked for English

Reviewer 3 Report
Comments and Suggestions for Authors
tRNAs contain many modifications including the sulfur substitution of an oxygen atom on the nucleobase (e.g., 2-thiouridine, S2U). The authors previously discovered that S2U underwent desulfurization under in vitro oxidative conditions. In this work, the authors investigated and identified the desulfuration of RS2U in natural tRNAs in live cells can be also occurred under oxidative stress. The authors employed a series of cells modes and used LC-MS/MS analysis to confirm the desulfuration productions. This research is interesting and important. The manuscript can be published after some minor revisions.
1. Under some pathological conditions, endogenous ROS levels may be high enough to induce desulfuration. The authors could add some comments for this point.
2. The cells will die partly under a certain ROS concentration; and the direct comparison of data from the treated cells may contain both live and death cells. Therefore, it is still not much clear that the desulfuration of tRNA could induce the cellular death.
3. The high IC50 values of NaOCl in Table 2 are unexpected. The NaOCl solution should be unstable, and it is better to test the concentration using the absorption at λ = 292 nm (ε = 350 M-1·cm-1).
4. They stated the LC-MS/MS experiments were repeated three times, and error bars should be added in the Figures.
5. Time-dependent cellular experiments may be useful to reveal more changes of 2-thiouridine desulfuration in HEK293 mutant variants.
Author Response
Response for Reviewer 3
We thank you for the positive evaluation of our manuscript, thank you for questions and valuable comments. All changes in the revised manuscript are highlighted in red. Below you will find our answers to individual questions and suggestions.
- Under some pathological conditions, endogenous ROS levels may be high enough to induce desulfuration. The authors could add some comments for this point.
As it is known from studies on oxidative stress (exogenous and/or endogenous, which occurs in the body in certain pathological disorders/diseases), free radicals, including reactive oxygen species (ROS), generated during oxidative stress are capable of altering the structure and damaging various biomolecules in the body that are essential for proper functioning. These include proteins, lipids, polysaccharides and also nucleic acids, including RNA and tRNA. Oxidatively damaged biomolecules impair cellular functions and contribute to the pathology of a variety of diseases. Our current studies on the oxidative damage of 2-thiouridine are based on previous observations that S2U is desulfurized in an oxidizing environment, resulting in products without a sulfur atom (H2U and U). These "in vitro" observations led us to the question whether a similar phenomenon could occur in living cells, which is particularly relevant in the context of tRNA containing 2-thiouridine derivatives in the wobble position of the anticodon. Therefore, we performed studies under cellular conditions, as we were aware that these are also model conditions. We show that under our experimental conditions, that mimic oxidative stress, we can observe to some extent a desulfuration of tRNA. We hypothesize that such an effect could be visible under certain pathological conditions in the body when endogenous ROS levels are high enough. When ROS levels are lower, such damage to tRNA may not occur, and in consequence might not be observed.
Fragments of the text written above are included in the abstract, introduction and conclusions of the publication.
- The cells will die partly under a certain ROS concentration; and the direct comparison of data from the treated cells may contain both live and death cells. Therefore, it is still not much clear that the desulfuration of tRNA could induce the cellular death.
Our studies are carried out under model conditions, in cells that have been exposed to oxidative stress during growth. To avoid direct effects of the oxidizing agent on tRNA, we determined the IC50 values for each oxidizing compound and for each cell type. We performed the desulfuration studies within the so-called "safe concentration" of the oxidant. Exceeding this concentration of oxidizing agent resulted in cell damage and direct oxidation of tRNA, which was simultaneously visible during LC-MS/MS analysis of nucleosides and manifested in the rapid increase in S2U desulfuration products (Figure 6, Figure 9, etc.). Interestingly, we were also able to identify S2U desulfuration products in so-called untreated cells, e.g. cancer cells, which have endogenous oxidative stress by nature (Figures 6-9). We do not assume that desulfuration causes cell death. The cell death observed at high ROS concentrations is caused by damage to multiple cellular biomolecules and the disruption of their function. The observed process triggered by oxidative stress in living cells can however IMPAIR the function of 2-thiouridine containing transfer RNAs, and alter the translation of genetic information (by for example lower translation dynamics).
- The high IC50 values of NaOCl in Table 2 are unexpected. The NaOCl solution should be unstable, and it is better to test the concentration using the absorption at λ = 292 nm (ε = 350 M-1·cm-1).
Sodium hydrochloride (NaClO) is a standard compound used in studies, e.g. on the effects of oxidative stress on tRNA damage (Chan CT, Dyavaiah M, DeMott MS, Taghizadeh K, Dedon PC, Begley TJ. A quantitative systems approach reveals dynamic control of tRNA modifications during cellular stress. PLoS Genet. 2010 Dec 16;6(12):e1001247). Studies on NaClO stability were described in a recently published article (Coaguila-Llerena H, Raphael da Silva L, Faria G. Research methods assessing sodium hypochlorite cytotoxicity: A scoping review. Heliyon. 2023 Nov 29;10(1):e23060. doi: 10.1016/j.heliyon.2023.e23060), which states that "...all solutions (NaClO) showed degradation versus time; however, this degradation occurred very slowly except for the group of solutions containing 5% available chlorine stored at 24°C. Solutions containing 0.5% available chlorine stored at 4°C and 24°C and 5% solutions stored at 4°C showed satisfactory stability at 200 days”. Furthermore, the concentrated reagent was stored at 4 °C and its solutions in were always freshly prepared at the appropriate concentrations immediately prior to administration to the cells. Therefore, we believe that our results and the concentration values reported in the manuscript are reliable.
- They stated the LC-MS/MS experiments were repeated three times, and error bars should be added in the Figures.
As stated in the publication, analysis of the nucleoside mixture from tRNA of cells exposed to oxidative stress was performed by LC-MS/MS method in two or three independent repetitions. The new figures with error bars have now been generated. The differences shown are not due to differences in the LC-MS/MS analysis results themselves, because the measurements are very precise, but to differences in the sample preparation (state of the cells, oxidation conditions, nuclease digestion, etc.).
- Time-dependent cellular experiments may be useful to reveal more changes of 2-thiouridine desulfuration in HEK293 mutant variants.
Thank you for this suggestion. Such experiments were performed on unmodified HEK293 cells and on some cancer cells derived from different cell lines (data not shown in the evaluated paper). Cells were exposed to oxidative stress induced by H2O2 (at the concentration below the IC50) for 0, 2, 4 or 6 hours. The results obtained depended on the individual features of the cells, but can be summarized as follows: during the first 2 hours of incubation, the greatest changes occur in the content of 5-substituted 2-thiouridine, there is the marked decrease in the content of this nucleoside with the simultaneous increase in the content of desulfuration products (mainly mcm5H2U). When the incubation time is prolonged, the most likely due to the effect of the cellular antioxidant system, the content of mcm5S2U stabilizes again or increases slightly after 4-6 hours, while at the same time the content of mcm5H2U decreases. The differences in the content of the second desulfuration product mcm5U were not so clear.
